# Development of a Multiplex Quantitative PCR for Detecting Porcine Epidemic Diarrhea Virus, Transmissible Gastroenteritis Virus, and Porcine Deltacoronavirus Simultaneously in China

**DOI:** 10.3390/vetsci10060402

**Published:** 2023-06-18

**Authors:** Jianpeng Chen, Rongchao Liu, Huaicheng Liu, Jing Chen, Xiaohan Li, Jianfeng Zhang, Bin Zhou

**Affiliations:** 1MOE Joint International Research Laboratory of Animal Health and Food Safety, College of Veterinary Medicine, Nanjing Agricultural University, Nanjing 210095, China; cjeng6752@163.com (J.C.); 2022807132@stu.njau.edu.cn (R.L.); 2021807200@stu.njau.edu.cn (H.L.); 2021207037@stu.njau.edu.cn (J.C.); fossy1012@163.com (X.L.); 2Institute of Animal Health, Guangdong Academy of Agricultural Sciences, Key Laboratory of Livestock Disease Prevention of Guangdong Province, Scientific Observation and Experiment Station of Veterinary Drugs and Diagnostic Techniques of Guangdong Province, Ministry of Agriculture and Rural Affairs, Guangzhou 510640, China; 13668939298@139.com

**Keywords:** multiplex quantitative PCR, porcine epidemic diarrhea virus, transmissible gastroenteritis virus, porcine deltacoronavirus, detection, mixed infection

## Abstract

**Simple Summary:**

For centuries, diarrhea disease has caused massive economic losses to the pig industry globally. Among RNA viruses causing pig diseases with the symptom of diarrhea include porcine epidemic diarrhea virus (PEDV), transmissible gastroenteritis virus (TGEV), and porcine deltacoronavirus (PDCoV), which all belong to the category of swine enteric coronaviruses and can result similar clinical symptoms in pigs, such as diarrhea, vomiting, dehydration, and so on. As a consequence, it is therefore necessary to develop a method that can detect and differentiate all three porcine enteric coronaviruses (PEDV, TGEV, and PDCoV) with a high sensitivity and specificity. In our study, we developed a multiplex quantitative PCR (qPCR) assay. We collected 462 samples of feces or small intestine from Jiangsu, Shandong, Hubei, Guangdong and Hunan provinces, following which the samples were detected for the evaluation of the application of the multiplex qPCR. The results indicated that the discrete positive rates of PEDV, TGEV, and PDCoV were 19.70%, 0.87%, and 10.17%, respectively. The mixed infection rates of PEDV/TGEV, PEDV/PDCoV, TGEV/PDCoV, and PEDV/TGEV/PDCoV were 3.25%, 23.16%, 0.22%, and 11.90%, respectively. The multiplex qPCR, a tool for differential and rapid diagnosing, can be put on the prevention and control of PEDV, TGEV, and PDCoV practically.

**Abstract:**

Porcine epidemic diarrhea virus (PEDV), transmissible gastroenteritis virus (TGEV), and porcine deltacoronavirus (PDCoV) belong to the category of swine enteric coronavirus that cause acute diarrhea in piglets, which has resulted in massive losses to the pig husbandry. Therefore, a sensitive and rapid detection method which can differentially detect these viruses that lead to mixed infections in clinical cases, is urgently needed. According to the conserved regions of the PEDV M gene, TGEV S gene, and PDCoV N gene, and the reference gene of porcine (β-Actin), we designed new specific primers and probes for the multiplex qPCR assay capable of simultaneously detecting three RNA viruses. This method, with a great specificity, did not cross-react with the common porcine virus. Moreover, the limit of detection of the method we developed could reach 10 copies/μL ,and the intra- and inter-group coefficients of variation of it below 3%. Applying this assay to detect 462 clinical samples which were collected in 2022–2023, indicated that the discrete positive rates of PEDV, TGEV, and PDCoV were 19.70%, 0.87%, and 10.17%, respectively. The mixed infection rates of PEDV/TGEV, PEDV/PDCoV, TGEV/PDCoV, and PEDV/TGEV/PDCoV were 3.25%, 23.16%, 0.22%, and 11.90%, respectively. All in all, the multiplex qPCR assay we developed as a tool for differential and rapid diagnosing can be put on the active prevention and control of PEDV, TGEV, and PDCoV, , which can create great value in the diagnosis of swine diarrhea diseases.

## 1. Introduction

Recently, porcine enteric viruses, which cause acute diarrhea in piglets, have led to huge economic losses in the swine industry in China [1]. Including porcine epidemic diarrhea virus (PEDV), transmissible gastroenteritis virus (TGEV), and porcine deltacoronavirus (PDCoV), porcine enteric viruses is thereby one of the most difficult problems in the pig husbandry all over the world [2]. Although pigs in all ages can be infected by porcine enteric viruses and show symptoms, acute diarrhea is especially severe in neonatal piglets leading to the high mortality rate, which can up to 100% [3,4]. PEDV, one of the *Alphacoronaviruses*, is an enveloped, single-stranded, positive-sense RNA virus, whose genome consists of a linear, single-stranded RNA molecule of 28 kb and is composed of seven open reading frames (ORFs) [5]. Transmissible gastroenteritis (TGE), a highly-contagious digestive tract disease, is caused by TGEV, which has a non-segmented, single- and positive- stranded RNA genome of 28.5 kb, and both ends of the genome are 5′-cap and 3′-poly (A) tail structures, respectively [6,7]. Furthermore, TGEV also belongs to the *Alphacoronavirus*. Different from PEDV and TGEV, PDCoV belongs to the *Deltacoronavirus* genus, and is an enveloped, single-stranded, positive-sense RNA virus with a genome of appropriately 25 kb in length [8]. As previously known, PEDV, TGEV, and PDCoV can cause piglets to experience similar clinical symptoms, such like vomiting, watery diarrhea, dehydration, and growth retardation [8,9,10], resulting in the difficult diagnosis in this field. Furthermore, lots of co-infection cases regarding porcine enteric coronavirus have been reported [11,12,13]. Therefore, it is of urgent need to develop a differential diagnosis for these three coronaviruses.

With the advantages of a good accuracy, high sensitivity, and high specificity, multiplex quantitative PCR (qPCR) is a highly efficient method as multiple viruses can be detected simultaneously in a single cube of reaction system, and analyze results directly without electrophoresis compared with PCR [14,15,16]. Furthermore, this method also benefits from greatly saving materials and testing time, particularly when we need to detect a lot of samples at the same time or perform the differential detection for mixing infections. Nowadays, multiplex quantitative qPCR assays for the simultaneous detection of PEDV, TGEV, and PDCoV have been reported. However, the assays that have been developed for the simultaneous detection of PEDV, TGEV, and PDCoV, do not encompass both ahigh sensitivity and specificity. In this study, we developed a multiplex qPCR assay for the differential detection of PEDV, TGEV, and PDCoV. 

## 2. Materials and Methods

### 2.1. Primers and Probes

To construct recombinant plasmid standards for qPCR, specific primers and probes were designed for the conserved regions of the PEDV M gene (AJ1102-R), TGEV S gene, (WH-1), and PDCoV N gene, respectively. Additionally, a control amplification was performed using β-Actin as an internal reference. The primers of TGEV S, PEDV M, and PDCoV N gene, and probes for the qPCR utilized in the study were synthesized by Genscript Biotech Co., Ltd. (Nanjing, China), and are listed in Table 1.

### 2.2. Standard Strains and Clinical Samples

Clinical samples that tested positive for PEDV, TGEV, CSFV, JEV, RVA, PRRSV, PCV2, PCV3, ASFV, and PRV, as confirmed by PCR, were stored in our lab. PDCoV strain was a gift from Researcher Zhang, Guangzhou Veterinary Research Institute (Guangzhou, China). The TGEV/PEDV vaccine consisted of the WH-1R strain + AJ1102-R strain. In the period of 2022–2023, a total of 462 clinical samples of feces and small intestines from pigs experiencing diarrhea issues were collected across various provinces. 

### 2.3. Extraction of RNA and Obtained cDNA 

We resuspended all 462 clinical samples, including feces and small intestines, in the solution of 0.9% stroke-physiological saline, and all samples were then homogenized. Following the manufacturer’s instructions, we used the Biomiga^®^ DNA/RNA Multiprep Kit (Biomiga, Hangzhou, China) for extracting RNA from the 150 μL homogenized sample. To reverse transcribe the extracted RNA into cDNA, the TransScript^®^ II Reverse Transcriptase Kit (TransScript, Beijing, China) was employed, in accordance with the manufacturer’s guidelines. We stored the resulting cDNA products in a −40 °C fridge. For the multiple qPCR analysis, the cDNA obtained from samples served as the template DNA, while the viruses were employed for specificity testing. 

### 2.4. Recombinant Plasmid Construction

The conserved regions of PEDV M, TGEV S, and PDCoV N were cloned into the pEasy vector and produced three recombinant plasmids, named pEasy-PEDV, pEasy-TGEV, and pEasy-PDCoV, respectively, which were sent to Tsingke Biotechnology Co., Ltd. (Beijing, China) for DNA sequencing. The concentration of the plasmid standards was determined using a NanoDrop spectrophotometer (Thermo Fisher, Waltham, MA, USA). Finally, the copy numbers of the three recombinant plasmids were calculated using the well-known formula [17,18].

### 2.5. Optimization of Conditions for Multiplex qPCR

As templates for amplification, the plasmid pEasy-PEDV, pEasy-TGEV, and pEasy-PDCoV were diluted 10-fold from 3 × 10^7^ copies to 3 × 10^4^ copies per microliter (μL), respectively. Alternatively, we also combined all diluted plasmids at the same proportion for a mixed standard plasmid concentration of 1 × 10^4^ copies/μL. For detection, four pairs of qPCR primers along with four probes, whose concentration was 10 μM were added to a single reaction cube. For optimizing the reaction conditions of the method that we have established, various additional volumes of the probes and primers were employed in order to determine the final concentration of each primer and probe. The reaction procedure involved an initial pre-denaturation step at 95 °C for 180 s, followed by 40 cycles consisting of denaturation at 95 °C for 10 s, annealing at 60 °C for 10 s, and extension at 72 °C for 20 s, respectively. The reaction mixtures included a 10 μL mix solution for qPCR which was purchased from Vazyme (Nanjing, China), with each primer being from 0.1 μL to 1.8 μL, each probe from 0.1 μL to 0.8 μL, and 2 μL of templates and ddH_2_O to form a final volume of 20 μL. This optimized reaction procedure of qPCR was determined by the instructions of the qPCR mix. The qPCR instrument was set to detect fluorescence signals in four channels, including FAM, HEX, Texas Red, and Cy5, respectively which could all be differentiated in our instrument. Lastly, we collected fluorescence signals using the qPCR instrument, which was purchased from IDEXX, Westbrook, ME, USA.

### 2.6. Standard Curves

We performed experiments using the optimal reaction condition, diluting recombinant plasmids ten-fold within a range of 10^5^ to 10^1^ copies per microliter (μL), respectively. Subsequently, these diluted plasmids were used as templates at seven different concentrations for qPCR. Utilizing Prism software, we generated standard curves by plotting the logarithm of the copy numbers against the corresponding Ct values, serving as the y-axis and x-axis, respectively. This visualization allowed us to analyze and interpret the data effectively. 

### 2.7. Specificity

The three generated plasmid standards served as positive controls, while samples of eight common porcine pathogens, including PCV2, PCV3, JEV, RVA, etc., were utilized as templates. The ddH_2_O was employed as a negative control, and the specificity was verified using multiplex qPCR under the optimal reaction system.

### 2.8. Sensitivity

All plasmids were diluted ten-fold within a range of 10^7^ to 10^0^ copies per microliter (μL), respectively, and the diluted plasmids were then combined to create concentration gradients. Subsequently, with the optimal reaction condition, multiplex qPCR was conducted. To determine the detection limits of this assay, three independent experiments were performed for each concentration, thereby ensuring that the replicates were performed accurately.

### 2.9. Repeatability

We diluted all plasmids by a 100-fold, ranging from 10^6^ copies/μL to 10^2^ copies/μL, respectively. Subsequently, the diluted plasmids of each concentration gradient were mixed together. With the optimal reaction condition, multiplex qPCR was then conducted. For each concentration gradient, three replicates were conducted, and the assays were performed every other week. The resulting data from different groups with their corresponding concentrations were compared, and the coefficients of variation were calculated within each group (intra-group) and between different groups (inter-group) to validate the reproducibility of the assay.

### 2.10. Clinical Sample Detection

We collected 462 samples of feces or small intestine from the Jiangsu, Shandong, Hubei, Guangdong, and Hunan provinces, following which the samples were then detected for the evaluation of the application of the multiplex qPCR. The RNA of 462 clinical samples were extracted using the Biomiga^®^ DNA/RNA Multiprep Kit. We used these three plasmids and ddH_2_O as positive controls and the negative control, respectively. With the optimized reaction condition, multiplex qPCR was conducted with the aim to determine whether each pathogen was positive. 

## 3. Results

### 3.1. Optimization of Conditions for Multiplex qPCR

We optimized the additional volumes of the primers and probes repeatedly, and then we got the optimal condition for our method (Table 2). Finally, the fluorescence channels of the qPCR instrument were set as follows: channel 1: FAM, channel 2: HEX, channel 3: CY5, and channel 4: Texas Red, respectively. This experiment applied Mic PCR real-time PCR instrument to collect fluorescence signals.

### 3.2. Standard Curves

All recombinant plasmid standards were diluted in a ten-fold gradient, ranging from 10^5^–10^1^ copies/μL, respectively. These diluted plasmids were then subjected to multiplex qPCR within the optimized reaction conditions. With the Ct value that we obtained, which was used as the vertical coordinate, and the plasmid concentration logarithm, which was served as the horizontal coordinate, we established standard curves for our multiplex qPCR successfully. Figure 1 illustrates these results, showing high correlation coefficients of R^2^ = 0.9948 for PEDV, R^2^ = 0.9987 for TGEV, and R^2^ = 0.9994 for PDCoV, respectively. These findings indicate a strong linear relationship between the standard curves of all standard plasmids which we constructed in our study, and their corresponding Ct values. 

### 3.3. Specificity

As shown in Figure 2, only the cDNA extracted from the samples which were diagnosed as positive for PEDV, TGEV, or PDCoV were able to be detected by the proposed method. Conversely, the DNA or cDNA, which were extracted from the other common porcine viruses were detected as negative with this method. Furthermore, the amplification of β-Actin indicated the absence of false positives in these samples. Based on these findings, it can therefore be concluded that this method exhibits an excellent specificity.

### 3.4. Sensitivity

To assess the sensitivity of the assay, PEDV, TGEV, and PDCoV were assessed using the optimal reaction system at seven different concentrations, ranging from 1 × 10^7^ copies/μL to 1 × 10^0^ copies/μL, respectively. As depicted in Figure 3, the assay demonstrated a detection limit of 10^1^ copies/μL for PEDV, TGEV, and PDCoV. These results indicate that the assay also exhibits an excellent sensitivity in detecting these pathogens.

### 3.5. Repeatability

The results, presented in Table 3, demonstrated that within the reproducibility tests for both the intra-group and inter-group, the Ct values were kept stable, as the coefficients of variation (CVs) largely ranged from 0.2% to 1.2%, and from 0.8% to 2.7%, respectively. These findings indicate that our method remained stable and reliable due to the repeatability test. 

### 3.6. Detection of the Clinical Samples

A total of 462 clinical samples were collected from various farms in the Jiangsu, Guangdong, Hubei, Shandong, and Hunan provinces between the years of 2022 and 2023, respectively. The cDNA of PEDV, TGEV, and PDCoV in these samples was detected using the established method. It is worth noting that the multiplex quantitative qPCR assay results were meticulously compared to the reference method in order to validate their accuracy and consistency. These comparative findings have been effectively presented in Table 4, highlighting the concordance between the multiplex qPCR assay and the reference method. Significantly, the agreement observed between these two approaches consistently exceeded 95%, thereby providing compelling evidence to support the notion that the established multiplex qPCR method possesses a substantial clinical significance and an unwavering reliability.

The meticulous comparison between the multiplex qPCR assay and the reference method allowed for a comprehensive assessment of their performance characteristics. The results, as shown in Figure 4, and in Table 4 and Appendix A, illustrated a remarkable level of agreement between these two techniques, reinforcing the credibility of the multiplex qPCR method. With the agreement consistently surpassing 95%, it became evident that the established approach not only demonstrates reliable and reproducible results, but also maintains a high degree of conformity with the reference method.

Moreover, according to the results displayed in Table 5, the positive rates for PEDV, TGEV, and PDCoV were found to be 19.70% (91/462), 0.87% (4/462), and 10.17% (47/462), respectively. The positive rates for the mixed infections of PEDV/TGEV, PEDV/PDCoV, TGEV/PDCoV, and PEDV/TGEV/PDCoV were 3.25% (15/462), 23.16% (107/462), 0.22% (1/462), and 11.90% (55/462), respectively. Detected by the specific primers and probe of β-Actin, all samples exhibited positive results, thereby confirming the proper sampling procedures without any false positives observed. Figure 5 and Table 6 present the distribution of the clinical samples.

## 4. Discussion

Coronaviruses, the longest genome virus discovered by humans to date, are named after their “crown-shaped” appearance under the electron microscope. As an RNA virus, it is more prone to genetic mutations and recombination to adapt to different environments and hosts. Therefore, it has many hosts in nature, and is also a pathogenic pathogen for many diseases, causing respiratory, digestive, and nervous system-related diseases in both mammals and birds [19].

Coronaviruses commonly affect pigs and are primarily associated with infectious diseases affecting the digestive system. Clinical manifestations primarily involve gastrointestinal symptoms, such as watery diarrhea, vomiting, dehydration, and weight loss. These viruses are collectively known as porcine enteric coronaviruses (PECs) due to the predominant tissue pathology occurring in the intestines. Examples of PECs include transmissible gastroenteritis virus (TGEV), porcine epidemic diarrhea virus (PEDV), porcine deltacoronavirus (PDCoV), and others. In 1935, TGE was first reported as a new porcine enteric disease, but at that time it was only known as an infectious disease that could cause digestive symptoms. It was only over a decade later that the pathogen responsible for TGEV was discovered in the United States and subsequently named TGE. In the winter of 1971, another new porcine acute enteric disease appeared in the UK, and its pathogen was named PEDV. In winter 2010, a highly virulent strain of PEDV GII emerged in South China, with an infection rate and death rate as high as 80%. This then spread to several countries in Asia, including Japan, Thailand, and so on, causing enormous economic losses to the pig industry in Asian countries. In 2012, cases related to PDCoV were first reported, and in 2014 an outbreak of PDCoV occurred in the United States, which about 30% of diarrhea-affected pigs diagnosed with PDCoV.

Porcine coronavirus has been ravaging the pig industry for decades, but its infection rate on farms remains high. Vaccination against TGEV and PEDV cannot provide a sufficient and effective protection, and there is still no effective vaccine available for PDCoV. To effectively prevent and control porcine coronavirus, strict biosecurity and production management are necessary, and timely, rapid, and accurate clinical diagnosis is the prerequisite for effective prevention and control. Due to the serious situation of mixed infections of porcine coronavirus, and the very similar clinical symptoms and pathological characteristics, it is not possible to make a differential diagnosis based on these factors alone. Therefore, it is necessary to develop a rapid, accurate, and supportive diagnostic test method for differentiation.

Several multiplex qPCR assays, which were established for the differential diagnoses among a few porcine viruses, have been published in recent years [14,15,20,21,22,23,24,25,26,27]. However, their methods have a low sensitivity, with the limit of detection (LOD) not being able to reach the 10 copies/μL for all of the pathogens. Considering PECs are too difficult to be purified in most farms, detection methods with a low sensitivity cannot provide timely disease monitoring for farms; or have a weak specificity, which cause the weak cross-reactivity with the other common pig pathogens. Inaccurate results from weakly specific methods also increase workload in clinical disease monitoring and undermine people’s confidence in the correct results.

In our developed multiplex RT-qPCR, prior to designing the primers and probes for qPCR, we searched strains of PECs which were prevalent in China in recent years on GenBank, compared them to the classical strains, and found highly conserved regions for designing the primers and probes. Eventually, we chose to design primers and probes on the PEDV M, TGEV S, and PDCoV N genes. Our detection method has unique characteristics and possesses several advantages over other detection methods due to these newly designed primers and probes. Firstly, the method has a high sensitivity. The newly designed qPCR primers and fluorescent probes enable the amplification efficiency of qPCR to reach over 90%, meaning the LOD for these three pathogens can reach 10 copies/μL (final reaction concentration of 1 copies/μL). We used 1 copies/μL template DNA in the sensitivity test, but all the results were negative. Although the Ct values of the samples was over 35 when the LOD reached 10 copies/μL, it was still found as clinically significant. When the Ct values were around 35, it meant that the viral load of the sample was at a relatively low level. We cannot solely rely on the Ct values for judgment; we also need to consider the clinical symptoms of the pig. The test results mainly serve as a reference. This method can timely monitor the occurrence of diseases in the early stage of PEC outbreak. Then, the feedback or antibody injection can be conducted in the early stages of an acute outbreak, which can thereby minimize economic losses in the farm. Secondly, the method has a strong specificity, which can specifically detect PEDV, TGEV, and PDCoV without cross-reacting with other common pig pathogens. The specific probes we designed avoided non-specific results, which were caused by the short-target fragments. Accurate detection results are crucial for disease monitoring in the farm. Thirdly, the method has a good repeatability, with the intra-group and inter-group variation coefficients mostly being around 1%, and the highest not exceeding 3%, which proves the stability and reliability of the detection results. The comparison of the clinical sample detection results with other detection methods shows a coincidence rate of about 95%.

Furthermore, non-standard sampling procedures can often lead to missed inspections. To address this issue, several established multiplex quantitative PCR methods have been reported, incorporating the detection of β-Actin, the internal reference gene, to prevent the incorrect outcomes. In the present study, the β-Actin gene was selected as an internal reference for amplification, aiming to minimize the occurrence of false-negative results. Therefore, the multiplex RT-qPCR assay that we established for PEDV, TGEV, and PDCoV, was also able to detect the porcine endogenous gene β-Actin, which was able to detect these three pathogens simultaneously, and ensure the accuracy and reliability of the results at the same time. The resulting data show that the established multiplex quantitative qPCR assay has a good reproducibility and sensitivity, as well as offering a rapid, convenient way for the differential detection of PEDV, TGEV, and PDCoV in pig farms.

The 462 samples obtained from five provinces in China between 2022–2023 were assessed by the developed assay. In addition, in order to co-verify the validity of the results, all 462 samples were also evaluated by the reference method, which revealed that the two methods had a coincidence rate of about 95%. The results showed that the positive rates of PEDV, TGEV, and PDCoV were 19.70% (91/462), 0.87% (4/462), and 10.17% (47/462), respectively. The positive rates of PEDV/TGEV, PEDV/PDCoV, TGEV/PDCoV, and PEDV/TGEV/PDCoV mixed infections were 3.25% (15/462), 23.16% (107/462), 0.22% (1/462), and 11.90% (55/462), respectively. Along with these data from our assay, the co-infected rates of PDCoV/PEDV and all three viruses were 25% and 20.77%, respectively, suggesting that the PEDV, TGEV, and PDCoV co-infections remained prevalent in a lot of pig farms. Recently, some reports showed that in China, common viral pathogens causing diarrhea include PEDV, TGEV, and PDCoV in a lot of pig farms, and PEDV was determined the most frequent driver compared to the other porcine viruses. Furthermore, the pigs were found to be most commonly co-infected by PEDV and PDCoV [16,27,28,29,30].

## 5. Conclusions

A successful establishment of a multiplex quantitative PCR (qPCR) assay has enabled the simultaneous detection of PEDV, TGEV, and PDCoV. This assay serves as a rapid and convenient way for the prompt identification of pig diarrhea, which is caused by porcine enteric coronaviruses (CoVs) under field conditions. The detection results also revealed significant instances of mixed infection, such as PEDV/PDCoV and PEDV/TGEV/PDCoV, within China. Therefore, in the prevention of swine diarrhea diseases, a crucial aspect is the implementation of differential diagnosis techniques.

## Figures and Tables

**Figure 1 vetsci-10-00402-f001:**
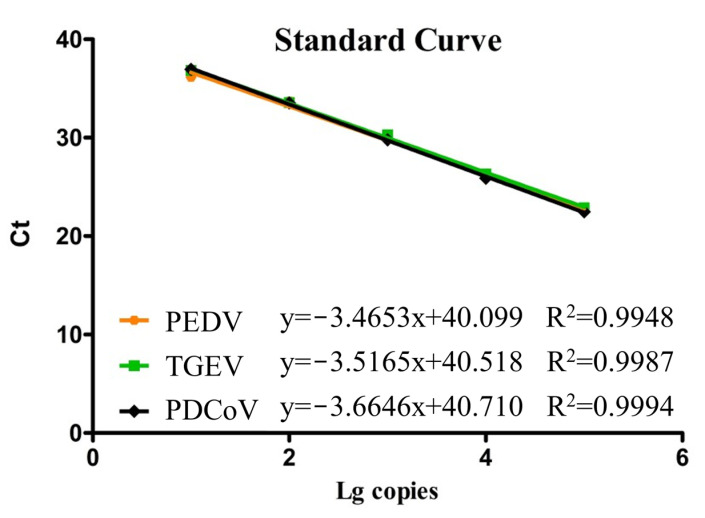
Standard curves of the multiplex qPCR assay.

**Figure 2 vetsci-10-00402-f002:**
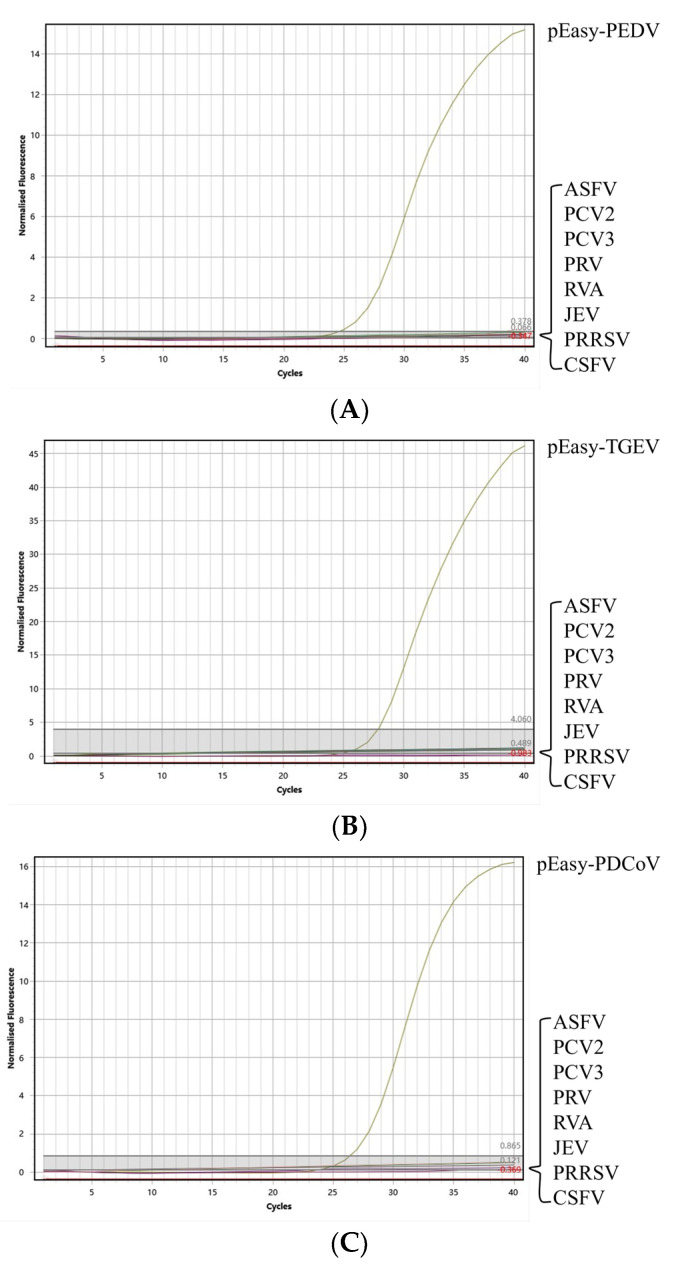
Amplified curves of the specificity test of the triplex quantitative PCR method. (**A**) Specificity test of PEDV; (**B**): specificity test of TGEV; and (**C**): specificity test of PDCoV.

**Figure 3 vetsci-10-00402-f003:**
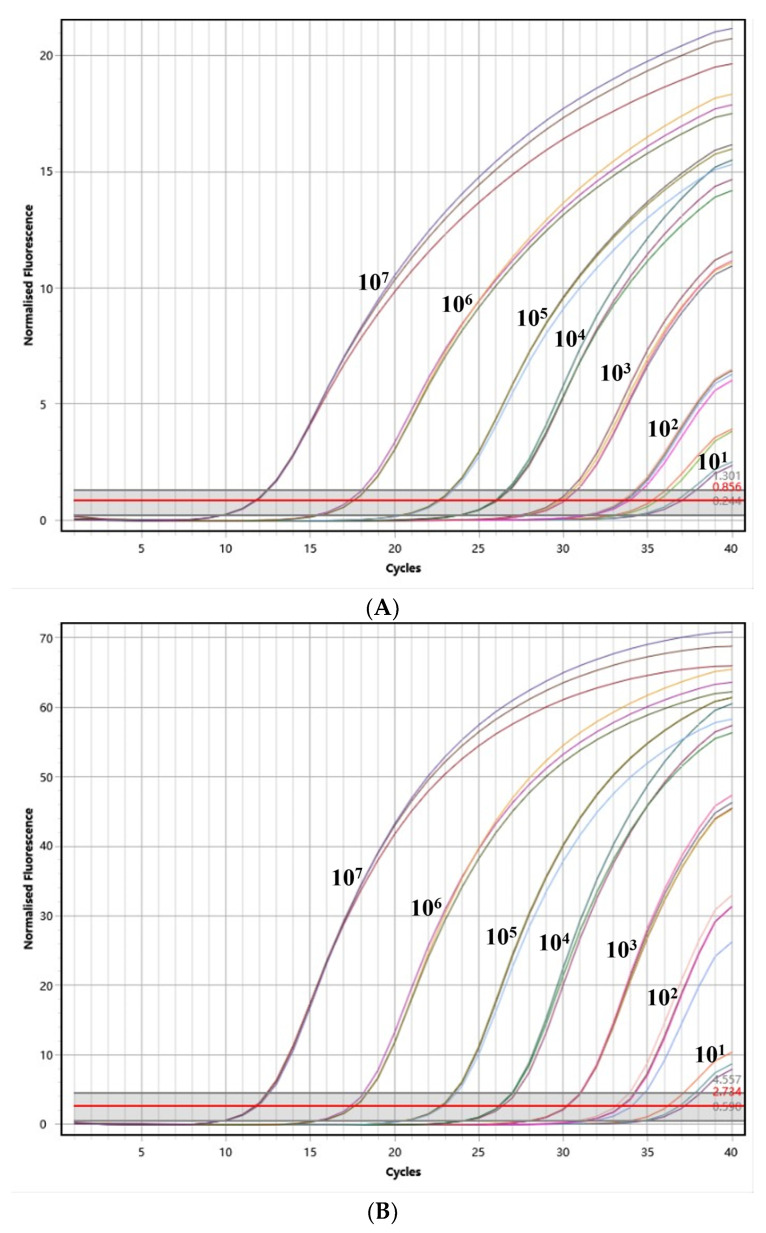
The test for the sensitivity of the triplex quantitative PCR method. (**A**) The test for the sensitivity of PEDV; (**B**) the test for the sensitivity of TGEV; and (**C**) the test for the sensitivity of PDCoV.

**Figure 4 vetsci-10-00402-f004:**
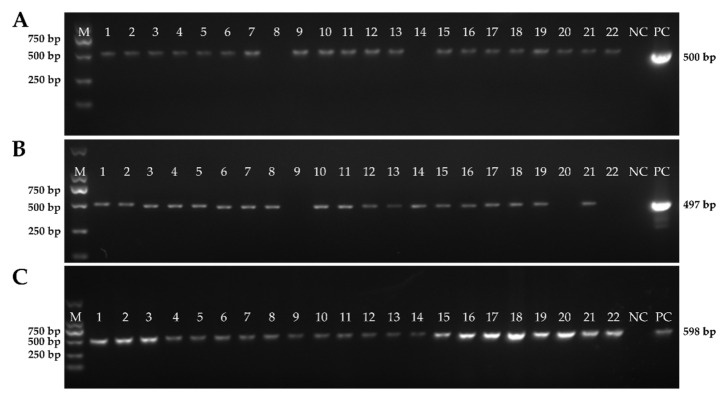
The detailed information of the clinical samples using PCR. (**A**) PCR product of samples for detecting PEDV; (**B**) PCR product of samples for detecting TGEV; and (**C**) PCR product of samples for detecting PDCoV.

**Figure 5 vetsci-10-00402-f005:**
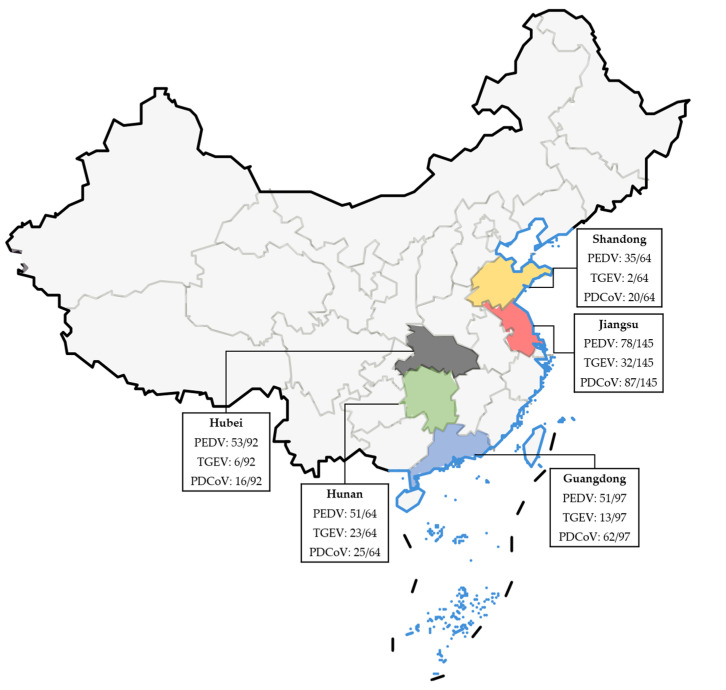
Epidemiological distribution maps of PEDV/TGEV/PDCoV in clinical samples from five provinces in China.

**Table 1 vetsci-10-00402-t001:** Primers and probes.

Primers and probes	Sequences (5′ end to 3′ end)	Length (bp)	Use
PEDV-F	CCCGTTGATGAGGTGATTG	500	Amplification of M
PEDV-R	TTGGCGACTGTGACGAAAT		
PEDV-qF	GACGCGCTTCTCACTACTTC	134	qPCR for the detection of M
PEDV-qR	TGTACGCCAGTAGCAACCTT		
PEDV-probe	FAM-TGCAGACCTGTCGGCCCATCA-BHQ1		
TGEV-F	GTCAACCCATAGCCTCAA	497	Amplification of S
TGEV-R	GCCACTAAGTAGCGTCCT		
TGEV-qF	ACATAGTGGGTGTACCGTCTG	140	qPCR for the detection of S
TGEV-qR	GCCACTAAGTAGCGTCCTGT		
TGEV-probe	CY5-AGCACTGACAAATCGTGCACACCA-BHQ2		
PDCoV-F	TACTCATCCTCAGTTTCGTG	598	Amplification of N
PDCoV-R	ACCCGTCTTCTCAGTGTCT		
PDCoV-qF	CAGTTTCGTGGCAATGGAGT	79	qPCR for the detection of N
PDCoV-qR	TGGTGTAACGCAGCCAGTAG		
PDCoV-probe	HEX-CCGCTTAACTCCGCCATCAAACCCG-BHQ1		
ACTB-qF	CCCTGGAGAAGAGCTACGAG	175	qPCR for the detection of β-Actin
ACTB-qR	AGGTCCTTCCTGATGTCCAC		
ACTB-probe	Texas Red-CGGCAACGAGCGCTTCCGGT-BHQ2		

**Table 2 vetsci-10-00402-t002:** The optimized multiplex qPCR reaction conditions.

Component	Volume (μL)
2 × AceQ qPCR probe master mix	10
TGEV-qF/qR (10 μM)	0.6 (0.3 μM)
TGEV-Probe (10 μM)	0.3 (0.15 μM)
PEDV-qF/qR (10 μM)	0.6 (0.3 μM)
PEDV-Probe (10 μM)	0.3 (0.15 μM)
PDCoV-qF/qR (10 μM)	0.6 (0.3 μM)
PDCoV-Probe (10 μM)	0.3 (0.15 μM)
Template DNA	2
ddH_2_O	Up to 20

**Table 3 vetsci-10-00402-t003:** Repeatability analysis for the triplex quantitative PCR method.

Plasmids	Concentration (Virus copies/μL)	Intra-group			Inter-group		
		Mean Ct	S.D.	CV (%)	Mean Ct	S.D.	CV (%)
PEDV	10^2^	32.355	0.189	0.6	32.759	0.286	0.8
	10^4^	25.208	0.042	0.2	25.386	0.242	0.9
	10^6^	16.418	0.193	1.2	16.969	0.431	2.5
TGEV	10^2^	32.063	0.183	0.6	32.576	0.381	1.2
	10^4^	24.719	0.058	0.2	25.309	0.429	1.7
	10^6^	16.421	0.167	1.0	17.047	0.452	2.7
PDCoV	10^2^	32.361	0.03	0.0009	32.526	0.133	0.004
	10^4^	25.176	0.005	0.0002	25.428	0.250	0.009
	10^6^	17.279	0.093	0.005	17.16	0.085	0.004

**Table 4 vetsci-10-00402-t004:** Agreement between the multiplex qPCR and the reference methods.

Detection Method	Number of Positive Samples		
	PEDV	TGEV	PDCoV
Multiplex qPCR	268	75	210
Reference methods	258	71	201
Agreements	96.27%	94.67%	95.71%

**Table 5 vetsci-10-00402-t005:** Clinical samples detection by the triplex quantitative PCR method.

Pathogens	Number of Positive Samples	Infection Rate (%)
PEDV	91	19.70
TGEV	4	0.87
PDCoV	47	10.17
PEDV/TGEV	15	3.25
PEDV/PDCoV	107	23.16
TGEV/PDCoV	1	0.22
PEDV/TGEV/PDCoV	55	11.90
β-Actin	462	100
In total	462	/

**Table 6 vetsci-10-00402-t006:** The detail information on clinical sample testing.

Province	Amount	Positive samples			Negative samples
		PEDV	TGEV	PDCoV	
Jiangsu	145	78	32	87	52
Guangdong	97	51	13	62	25
Hubei	92	53	6	16	34
Shandong	64	35	2	20	28
Hunan	64	51	23	25	3
Sum	462	268	76	210	142

## Data Availability

Not applicable.

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
