# Peer review of "Development of a Multiplex Quantitative PCR for Detecting Porcine Epidemic Diarrhea Virus, Transmissible Gastroenteritis Virus, and Porcine Deltacoronavirus Simultaneously in China"

_vetsci, 2023, doi:10.3390/vetsci10060402_

Round 1
Reviewer 1 Report
In this study, a multiplex RT-qPCR method was established for the diagnosis of TGEV, PEDV, and PDCoV. These three porcine enteric coronaviruses (PECs) pose challenges in clinical differentiation and have high rates of co-infection. The author redesigned three pairs of specific fluorescent quantitative primers and probes, making the method highly specific for the detection of TGEV, PEDV, and PDCoV. The results for other common porcine pathogens were negative. Furthermore, the method exhibited strong sensitivity, with the limit of detection of 10 copies/μL for each of the three pathogens. The detection results were stable and reproducible, with intra- and inter-assay coefficients of variation not exceeding 3%. Moreover, the method allowed simultaneous detection of TGEV, PEDV, and PDCoV in a single system, using different channels that did not interfere with each other.
The main issues addressed in this study are as follows:
(1) Based on the standard curve, the Ct values exceeding 35 when the limit of detection reach 10 copies/μL. However, it is uncertain whether these results have clinical values. Please explain it.
(2) The specific primers for PDCoV were used to amplify a target fragment of only 79 bp. This raised the concern of potential non-specific amplification, which should be discussed in the section.
(3) What are the initial concentrations of primers and probes? Please specify it, not just the amount added.
(4) In Sensitivity, have you used template DNA of which concentration less than 10 copies/μL, making sure that limit of detection just reached 10 copies/μL?
(5) These are some writing errors, like “ddH2O” in 2.10. Clinical sample detection, which should be “ddH2O”. Please correct it all.
(6) How did you determine the reaction procedure, which is optimized for the qPCR?
(7) In Specificity, why those eight pathogens were chosen as templates?
In this study, a multiplex RT-qPCR method was established for the diagnosis of TGEV, PEDV, and PDCoV. These three porcine enteric coronaviruses (PECs) pose challenges in clinical differentiation and have high rates of co-infection. The author redesigned three pairs of specific fluorescent quantitative primers and probes, making the method highly specific for the detection of TGEV, PEDV, and PDCoV. The results for other common porcine pathogens were negative. Furthermore, the method exhibited strong sensitivity, with the limit of detection of 10 copies/μL for each of the three pathogens. The detection results were stable and reproducible, with intra- and inter-assay coefficients of variation not exceeding 3%. Moreover, the method allowed simultaneous detection of TGEV, PEDV, and PDCoV in a single system, using different channels that did not interfere with each other.
The main issues addressed in this study are as follows:
(1) Based on the standard curve, the Ct values exceeding 35 when the limit of detection reach 10 copies/μL. However, it is uncertain whether these results have clinical values. Please explain it.
(2) The specific primers for PDCoV were used to amplify a target fragment of only 79 bp. This raised the concern of potential non-specific amplification, which should be discussed in the section.
(3) What are the initial concentrations of primers and probes? Please specify it, not just the amount added.
(4) In Sensitivity, have you used template DNA of which concentration less than 10 copies/μL, making sure that limit of detection just reached 10 copies/μL?
(5) These are some writing errors, like “ddH2O” in 2.10. Clinical sample detection, which should be “ddH2O”. Please correct it all.
(6) How did you determine the reaction procedure, which is optimized for the qPCR?
(7) In Specificity, why those eight pathogens were chosen as templates?
Author Response
Response:
(1) It still has clinical values. When the Ct values around 35, it meant that the viral load of the sample was at a relatively low level. We cannot solely rely on the Ct values for judgment; we also need to consider the clinical symptoms of the pig. The test results mainly serve as a reference.
(2) In qPCR tests, the target fragment length is generally required to be within 200bp because longer fragments would reduce efficiency. The specificity fluorescent probe we designed can avoid nonspecific results.
(3) The initial concentrations of primers and probes are both 10 mM. And the final concentration of primers and probes in the reaction cube are 0.3 mM and 0.15 mM respectively. We have updated the Table 2.
(4) Yes, we did use 1 copy/μL template DNA in the sensitivity test, but all the results were negative. Therefore, it was not mentioned in the figure and article.
(5) We have corrected it all.
(6) The optimized reaction procedure of qPCR is determined by the qPCR Mix. So we just followed the instruction
(7) Because those eight viruses are common porcine pathogens, if we can ensure that the method does not cross-react with those viruses, then we can ensure the specificity of the method.
Reviewer 2 Report
Chen Jianpeng et al., described the development of a multiplex quantitative PCR for detecting porcine epidemic diarrhea virus, transmissible gastroenteritis virus and porcine deltacoronavirus simultaneously in China in 2022-2023. All in all, the multiplex qPCR assay we developed. Detection of PEDV, TGEV and PDCoV is indeed important for the prevention and control of PEDV, TGEV and PDCoV, which can create great value in the diagnosis of swine diarrhea diseases. The authors' findings support the conclusion. The finding is interesting.
1. The title is too long. I would suggest that the title should be simpler. “in 2022-2023” in the title should be deleted.
2. The authors should check the grammar and formatting carefully.
The authors should check the grammar
Author Response
- The title is too long. I would suggest that the title should be simpler. “in 2022-2023” in the title should be deleted.
Response: Sure, we have already deleted “in 2022-2023” from the title.
- The authors should check the grammar and formatting carefully.
Response: We would check the grammar and formatting carefully again. Thank you for the reminder.
Reviewer 3 Report
General Comment: Why the reference model Beta -Actin was chosen as a control, please explain.
Also, please include the PCR product figure of each sample after PCR.
Author Response
- Why the reference model Beta -Actin was chosen as a control, please explain.
Response: Because Beta-Actin, an internal reference gene, is present in large quantities in samples from porcine. Chosen as a control, Beta-Actin can largely ensure that the samples belong to swine and guarantee the accuracy of positive results.
- Also, please include the PCR product figure of each sample after PCR.
Response: Thank you for the reviewer's suggestion. In the main body of the manuscript, we added the Figure. 4 that showed the PCR results of agarose gel electrophoresis (some samples). Alternatively, we have added a table (Table S1) as an attachment to record the PCR results of each sample in detail. Please check it.